# Impact of Scion and Rootstock Seedling Quality Selection on the Vigor of Watermelon–Interspecific Squash Grafted Seedlings

**Filippos Bantis [1,\*], Athanasios Koukounaras [1], Anastasios S. Siomos [1] and Christodoulos Dangitsis [2]**

[1] Department of Horticulture, Aristotle University, 54124 Thessaloniki, Greece; thankou@agro.auth.gr (A.K.); siomos@agro.auth.gr (A.S.S.)

[2] Agris S.A., Kleidi, 59300 Imathia, Greece; cdaggitsis@agris.gr

\* Correspondence: fbantis@agro.auth.gr; Tel.: +30-2310-994123

**Abstract:** Watermelon is mainly grafted onto interspecific squash, and during the season of high demand, seedlings of variable quality are used to cover grafting needs. The objective was to combine watermelon and interspecific squash of different seedling quality categories in order to obtain the optimal combination for the production of high-quality grafted watermelon seedlings. Acceptable seedlings of both species were grouped into quality categories, namely "low", "optimum", and "high". Seedlings of each quality category were combined with each other and grafted seedlings from the nine derived categories were evaluated at two time intervals, at 7 (exit from the healing chamber) and 14 (final product) days after grafting. At both time intervals, watermelon "high" combined with interspecific squash "optimum" exhibited relatively higher shoot length, stem diameter, leaf area, and shoot and root dry weight, as well as shoot dry weight-to-length ratio, which is a quality index. The study shows that watermelon scions should have "high" quality, while interspecific squash rootstocks should have "optimum" quality during grafting for the production of high-quality grafted plants. If possible, "low" to-be-grafted watermelon seedlings should be avoided because the grafted derived seedlings are considered low quality.

**Keywords:** *Citrullus lanatus*; *Cucurbita maxima* × *C. moschata*; TZ-148; vegetable grafting; quality indicators

## 1. Introduction

The production of watermelon is economically valuable on a worldwide scale. For its production, watermelon hybrids (scions) are mainly grafted onto cucurbit rootstocks, such as interspecific squash or bottle gourd. Grafting has desirable traits for growers, including plant vigor enhancement, growth uniformity, and earliness, as well as yield increase and the extension of the harvesting period, among others [1–4]. One of the most commonly employed rootstocks for watermelon throughout the world, TZ-148, is an interspecific squash hybrid (*Cucurbita maxima* × *C. moschata*), which provides the plant with the abovementioned traits. Nowadays, grafted watermelon seedlings are used in high percentages around the world. Over 90% of the watermelon cultivated area is planted with grafted seedlings in Greece (Th. Koufakis, Agris S.A., Kleidi, Imathia, Greece, personal communication), Japan, and South Korea, while in China, that number reaches 20% [1]. Grafted watermelon seedlings are produced in three separate stages: growing to-be-grafted seedlings (scion and rootstock) (stage I), the grafting and healing of grafted seedlings (stage II), and the acclimatization of grafted seedlings (stage III). Important parameters for successful grafting are the timing and proper connection between

the scion and rootstock parts during healing in order to form a healthy and vigorous grafted seedling. After the attachment, the callus starts differentiating into vascular cells and the procedure depends on environmental conditions, including temperature and relative humidity.

The use of high-quality grafted seedlings is a prerequisite for successful crop establishment and vegetable seedling nurseries are rapidly expanding [5]. However, modern nurseries face the problem of limited space available for cultivation due to high product demand during particular seasons for each species. Particularly for the production of each grafted watermelon seedling, two seedlings have to be cultivated, a watermelon scion and a rootstock seedling, in our case, interspecific squash. Therefore, the nursery is usually occupied by grafted watermelon seedlings as well as by scion and rootstock seedlings, which are used in another grafting batch. It is highly essential that scion and rootstock seedlings are of high quality at the time of grafting in order to produce greatly valued grafted seedlings. Recently, Bantis et al. [6] established objective criteria for the determination of watermelon and interspecific squash quality which may assist nurseries to select seedlings of certain quality characteristics and subsequently to enhance grafted seedling production. Moreover, a number of research groups [7,8] highlighted the importance of scion–rootstock hybrid interactions for the quality of watermelon. However, the literature lacks information about proper scion–rootstock seedling quality combinations for increasing the quality of grafted watermelon seedlings.

Grafted watermelon seedlings produced during the season of high demand have questionable quality since the available space for cultivation is limited and the nurseries are pressed for time. The objective of the present study was to test quality category combinations between watermelon scions and interspecific squash rootstocks in order to determine the obtained quality of the produced grafted seedlings, as well as to enhance the compatibility between the scion and rootstock by identifying their optimum combination. The gained information will be useful for nurseries and growers in order to facilitate production programming.

## 2. Materials and Methods

### 2.1. Plant Material and Growth Conditions

The experiment was conducted at a commercial high-tech nursery (Agris S.A. in Kleidi, Imathia, Greece) where standard commercial practices were applied. Measurements were performed at the laboratory of Vegetable Crops of Aristotle University of Thessaloniki.

Watermelon (*Citrullus lanatus*) "Celine F1" and interspecific squash (*Cucurbita maxima* × *C. moschata*) "TZ-148" (seeds of both species were provided by HM.Clause SA, Portes-Les-Valence, France) were used as scion and rootstock material, respectively. Seeds of watermelon and interspecific squash were sown in 171- and 128-cell plug trays (67 × 33 cm, G.K. Rizakos S.A., Lamia, Greece), respectively, which were filled with a mixture of peat, perlite, and vermiculite (5:1:2). Afterwards, the plug trays were placed for 72 (watermelon) or 48 (interspecific squash) h in a custom-built growth chamber under set conditions at 25 °C temperature, 95–98% relative humidity, and darkness until germination. Then, both scion and rootstock trays were moved to different compartments of a modern Venlo-type gable greenhouse. Scion trays were placed under 21.5 °C minimum night temperature (controlled with overground heating system) and under supplemental artificial lighting emitted by high-pressure sodium (HPS) lamps (MASTER GreenPower 600 W 400 V E40, Philips Lighting, Eindhoven, The Netherlands) at $100 \pm 10$ μmol m$^{-2}$ s$^{-1}$ and an 18 h photoperiod. Rootstock trays were placed under natural light for 7 days at 20 °C minimum night temperature followed by 3 days at 14 °C minimum night temperature in order to reduce the seedlings' growth rate and increase stem thickness.

### 2.2. Quality Category Combinations between Watermelon and Interspecific Squash Seedlings

Depending on their quality, watermelon and interspecific squash seedlings were grouped into categories, namely "low" (i.e., more compact than optimum quality), "optimum", and "high" (i.e., more developed than optimum quality) which were clearly distinguished by a number of

quantifiable parameters [6]. Briefly, for each watermelon quality category, leaf area ranges were 6–8, 8–10, and 10–12 cm$^2$, respectively, while shoot dry weights were about 0.07, 0.08, and 0.09 g, respectively. For every interspecific squash quality category, leaf area ranges were 1–3, 3–5, and over 5 cm$^2$ respectively, while shoot dry weights were about 0.17, 0.18, and over 0.19 g, respectively. The grafting combinations between quality categories from both species are depicted in Table 1. In order to adequately cover every quality category combination and obtain sufficient seedlings from each quality category, seedlings of both species were sown on three consecutive days but they were all grafted at the same time. Specifically, watermelon "low", "optimum", and "high" seedlings were grown for a total of 12, 13, or 14 days, respectively, until grafting, while interspecific squash seedlings were grown for a total of 11, 12, or 13 days, respectively, until grafting. Therefore, watermelon seedlings grown for 14 days were generally grouped in a higher quality category (e.g., high) compared to seedlings grown for 12 or 13 days (e.g., optimum or low), and the same applied for interspecific squash seedlings.

**Table 1.** Grafting combinations between quality categories of watermelon and interspecific squash seedlings.

| Abbreviation | Watermelon Quality Category | | Interspecific Squash Quality Category |
|:---:|:---:|:---:|:---:|
| WH × SH | High | × | High |
| WO × SH | Optimum | × | High |
| WL × SH | Low | × | High |
| WH × SO | High | × | Optimum |
| WO × SO | Optimum | × | Optimum |
| WL × SO | Low | × | Optimum |
| WH × SL | High | × | Low |
| WO × SL | Optimum | × | Low |
| WL × SL | Low | × | Low |

*2.3. Grafting, Healing, and Acclimatization of Grafted Seedlings*

After 12–14 and 11–13 days of growth for scions and rootstocks (at the stage of one true leaf), respectively, watermelon was grafted onto interspecific squash using the "splice grafting" technique. The two segments were held together by a silicon clip and the grafted seedlings were planted in 72-cell plug trays (50 × 30 cm, G.K. Rizakos S.A., Lamia, Greece). In addition, the rootstock's root system was completely cut off in order to increase grafting efficiency [9]. The trays were filled with a 3:1:1 mixture of peat, perlite, and vermiculite, including fertilizer (permanent NPK with Mg and trace elements; Osmocote Exact Mini, Geldermalsen, The Netherlands). In order to avoid critical errors, grafting was performed by experienced personnel.

Immediately after grafting, seedlings were placed on carriage (2.00 × 1.66 × 0.76 m) shelves in a healing chamber where they stayed for 7 days at 25 °C, with recirculating air, and high relative humidity (98% for the first four days, 93% for day five, and 89% for days six and seven). High relative humidity was absolutely necessary to prevent water loss and leaf dehydration. Light was provided by fluorescent tubes (Fluora 58W, Osram, GmbH, Munich, Germany) installed 30 cm from the plant tops, and emitting 45 µmol m$^{-2}$ s$^{-1}$ with a 16 h photoperiod for the first two days and 18 h for the following days. The photoperiod shift took place because seven days of an 18 h photoperiod proved stressful for the grafted seedlings, while 16 h for the first two days enhanced the vascular connection and new root development. Conditions were continuously monitored by a climate control system (Priva SA, De Lier, The Netherlands).

After exiting the healing chamber, the trays were moved to a greenhouse where they stayed for 7 days at 21.5 °C minimum night temperature. Artificial lighting was emitted by HPS lamps with an 18 h photoperiod and 60 ± 10 µmol m$^{-2}$ s$^{-1}$. The experiment was performed two times and similar conclusions were reached from both replications, thus, results from only one replication are presented.

*2.4. Sampling, Quality Categorization, and Measurements of Grafted Seedlings*

The sampling of grafted watermelon seedlings was performed at two time intervals, i.e., after exiting the healing chamber (7 days from grafting), and after one week of acclimatization in the greenhouse (14 days from grafting) when seedlings were labeled as "final product" and their quality evaluation was essential in order to determine the product marketability. Specifically, 12 evenly distributed seedlings per combination were sampled.

At each time interval, the grafted seedlings were grouped into quality categories, namely "not acceptable" (i.e., not marketable), "acceptable" (i.e., marketable), and "optimum" (obtained from [10]), assisted by experienced personnel. It should be noted that the "optimum" quality category of grafted seedlings was distinguished from the "optimum" of watermelon and to-be-grafted interspecific squash seedlings. Briefly, 7 days from grafting, the leaf area of each quality category was measured as below 20, 20–25, and over 25 cm$^2$, respectively, while shoot dry weight was below 0.17, 0.17–0.20, and over 0.20 g, respectively. Moreover, 14 days from grafting, the leaf area of each quality category was measured as below 45, 45–50, and over 50 cm$^2$, respectively, while shoot dry weight was below 0.30, 0.30–0.35, and over 0.35 g, respectively. Moreover, a number of morphological and growth parameters were evaluated, which proved valuable for classification into the abovementioned quality categories. Specifically, shoot length, stem diameter, and leaf thickness were determined using a digital caliper (Powerfix, Milomex, Pulloxhill, UK). Leaf area was measured using a leaf area meter (LI-3000C, LI-COR biosciences, Lincoln, Dearborn, MI, USA). Relative chlorophyll content was determined by a chlorophyll meter (CCM-200 plus, Opti-Sciences, Hudson, NH, USA), while leaf colorimetric parameters were obtained using a digital colorimeter (CR-400 Chroma Meter, Konica Minolta Inc., Tokyo, Japan). Shoot and root dry weights were measured after drying for three days in an oven. In addition, quality indices such as root-to-shoot (R/S) ratio and shoot dry weight-to-length (DW/L) were estimated.

*2.5. Statistical Analysis*

Statistical analysis was performed by IBM SPSS software (SPSS 23.0, IBM Corp., Armonk, NY, USA). Analysis of variance (ANOVA) included factorial and bifactorial tests, while mean comparisons were performed using Tukey's test at a significance level of $\alpha = 0.05$. Each mean value was computed from $n = 12$ seedlings. Normality was determined with a Kolmogorov–Smirnov test, while homogeneity of variances was determined with Levene's test at a significance level of $\alpha = 0.05$.

## 3. Results and Discussion

The quality of grafted seedlings is highly dependable on the scion and rootstock status as well as on their potential compatibility before grafting. Moreover, increased scion × rootstock compatibility during grafting leads to the production of high-quality grafted seedlings with better vascular connection and helps producers to better estimate their production quality and timing. Until recently, watermelon (scion), interspecific squash (rootstock), and grafted watermelon seedling quality was rather vague and defined by experience. Therefore, the producers did not have an objective way of defining seedling quality. However, important parameters for the quality characterization of the abovementioned seedlings have recently been properly determined [10].

Substantial visual quality differences were observed in both time intervals. According to the above, after exiting the healing chamber (7 days from grafting), over 80% of the seedlings from WH combinations were characterized as "optimum", whereas seedlings from WL combinations had the least "optimum" and the most "not acceptable" seedlings (Figure 1). Moreover, regardless the interspecific squash category, the grafted seedling quality deteriorated, from the watermelon category "high" (70–80% "optimum" grafted seedlings) to "optimum" (40–60% "optimum" grafted seedlings) and "low" (10–20% "optimum" grafted seedlings). At the second time interval, 14 days after grafting,

WH × SO had the most "optimum" seedlings, while WL combinations led to inferior seedlings, with over 50% "not acceptable" and 0% "optimum" seedlings regardless of the rootstock quality (Figure 1).

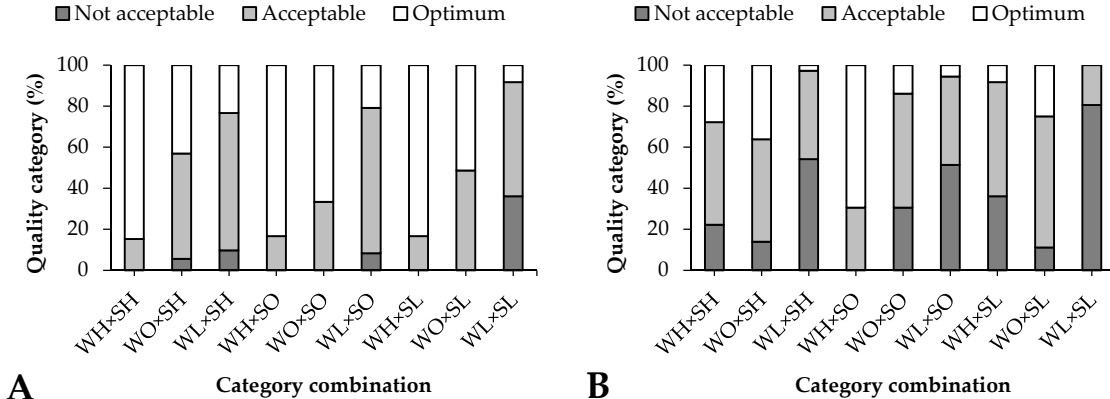

**Figure 1.** Quality categories (%) of grafted watermelon seedlings from nine quality category combinations obtained 7 (**A**) or 14 (**B**) days after grafting, as described in Table 1. Each percentage was computed from three trays each having *n* = 72 seedlings.

Color is a useful parameter in vegetable production as a product quality indicator [11]. Seven days from grafting, relative chlorophyll content was similar in all combinations, whereas colorimetry revealed differences in the seedlings' lightness (L*) and chroma (C*) parameters (Table 2). Bifactorial analysis showed that WL combinations led to seedlings with different L*, C*, hue angle (h°), and a*/b* (a*: red/green coordinate; b*: yellow/blue coordinate) values compared to the rest of the combinations, while the interspecific squash quality category did not impose a significant effect (Table S1). At the second time interval, 14 days after grafting, relative chlorophyll content was greater in WL combinations, while the tested colorimetric parameters were also variably affected by "low" watermelon scions (Table 3, Table S2). A strong linear relationship ($r^2 > 0.93$) has been reported between chlorophyll content (measured with soil plant analysis development-SPAD) and L* value [12]. Therefore, the darker color (lower L*) of WL combinations can be associated with a higher chlorophyll concentration in the leaves.

**Table 2.** Colorimetric parameters and relative chlorophyll content of grafted watermelon seedlings from nine quality category combinations obtained 7 days after grafting, as described in Table 1.

| Category Combination | Colorimetric Parameter | | | | Relative chl. Content |
|---|---|---|---|---|---|
| | **L*** | **C*** | **h°** | **a*/b*** | |
| WH × SH | 39.41 ± 0.31 bc | 19.25 ± 0.97 ab | 131.2 ± 0.4 a | −0.88 ± 0.01 a | 36.42 ± 2.72 a |
| WO × SH | 38.80 ± 0.23 bc | 17.90 ± 0.60 b | 131.4 ± 0.3 a | −0.88 ± 0.01 a | 43.78 ± 2.04 a |
| WL × SH | 40.37 ± 0.33 ab | 20.98 ± 0.83 a | 130.2 ± 0.4 a | −0.85 ± 0.01 a | 42.11 ± 3.28 a |
| WH × SO | 39.14 ± 0.2 bc | 19.35 ± 0.43 ab | 131.5 ± 0.2 a | −0.89 ± 0.01 a | 38.67 ± 1.24 a |
| WO × SO | 38.71 ± 0.43 bc | 19.08 ± 0.56 ab | 131.5 ± 0.3 a | −0.88 ± 0.01 a | 40.36 ± 1.82 a |
| WL × SO | 40.09 ± 0.34 abc | 19.82 ± 0.34 ab | 131.2 ± 0.3 a | −0.88 ± 0.01 a | 40.75 ± 3.89 a |
| WH × SL | 39.21 ± 0.36 bc | 19.58 ± 0.48 ab | 131.3 ± 0.2 a | −0.88 ± 0.01 a | 39.31 ± 1.34 a |
| WO × SL | 38.52 ± 0.56 c | 20.33 ± 0.48 ab | 131.0 ± 0.3 a | −0.87 ± 0.01 a | 38.96 ± 1.91 a |
| WL × SL | 41.19 ± 0.45 a | 21.65 ± 0.44 a | 130.2 ± 0.2 a | −0.85 ± 0.01 a | 36.48 ± 0.91 a |

Mean values (± SE) followed by different letters within a column are significantly different (α ≤ 0.05). Each mean value was computed from *n* = 12 seedlings. L*: lighting; C*: chroma; h°: hue angle; a*: red/green coordinate; b*: yellow/blue coordinate.

**Table 3.** Colorimetric parameters and relative chlorophyll content of grafted watermelon seedlings from nine quality category combinations obtained 14 days after grafting, as described in Table 1.

| Category Combination | Colorimetric Parameter | | | | Relative chl. Content |
|---|---|---|---|---|---|
| | L* | C* | h° | a*/b* | |
| WH × SH | 38.96 ± 0.27 a | 16.76 ± 0.59 a | 131.6 ± 0.3 c | −0.89 ± 0.01 a | 39.61 ± 2.04 b |
| WO × SH | 37.16 ± 0.40 b | 14.71 ± 0.39 abc | 132.8 ± 0.3 abc | −0.93 ± 0.01 abc | 44.59 ± 1.80 b |
| WL × SH | 37.07 ± 0.31 b | 13.25 ± 0.39 c | 134.3 ± 0.4 a | −0.97 ± 0.01 c | 47.66 ± 2.55 ab |
| WH × SO | 37.62 ± 0.39 ab | 14.78 ± 0.64 abc | 133.1 ± 0.2 abc | −0.94 ± 0.01 abc | 48.82 ± 3.17 ab |
| WO × SO | 37.76 ± 0.33 ab | 15.28 ± 0.30 abc | 132.6 ± 0.2 abc | −0.92 ± 0.01 abc | 42.60 ± 1.36 b |
| WL × SO | 37.35 ± 0.43 ab | 13.78 ± 0.66 bc | 133.6 ± 0.7 abc | −0.95 ± 0.02 abc | 49.00 ± 4.25 ab |
| WH × SL | 38.04 ± 0.40 ab | 16.39 ± 0.79 ab | 132.5 ± 0.5 abc | −0.92 ± 0.01 abc | 41.11 ± 2.20 b |
| WO × SL | 38.75 ± 0.58 ab | 16.88 ± 0.96 a | 132.1 ± 0.7 bc | −0.90 ± 0.02 ab | 42.21 ± 2.63 b |
| WL × SL | 37.31 ± 0.32 ab | 13.59 ± 0.57 c | 134.0 ± 0.5 ab | −0.97 ± 0.02 bc | 57.46 ± 2.29 a |

Mean values (± SE) followed by different letters within a column are significantly different ($\alpha \leq 0.05$). Each mean value was computed from $n$ = 12 seedlings. L*: lighting; C*: chroma; h°: hue angle; a*: red/green coordinate; b*: yellow/blue coordinate.

In order for the watermelon hybrid to benefit from the rootstock's advantages, a long shoot of the grafted seedling is highly important to avoid the ground, especially during transplantation. Height is considered an important quality indicator for many species since it is correlated with seedling vigor after transplantation [13]. Moreover, a proper vascular connection is the most crucial factor for the successful development of high-quality grafted seedlings. Seven days from grafting, shoot length was significantly greater in WH × SO (+15–18%) compared to WL × SO, WO × SL and WL × SL, whereas stem diameter was not significantly affected by the different combinations (Figure 2). Quite similarly, at 14 days from grafting, WH × SO developed significantly longer shoots (+13–21%) compared to WL × SO and WL × SL (Figure 3). In general, SL combinations produced the shortest shoots among the rest of the combinations (Table S2). In the meantime, stem diameter revealed significantly lower values in SH combinations (Figure 3, Table S2) even though SH had a rather thick stem before grafting [6]. Stem diameter is considered an efficient parameter for distinguishing grafted seedlings' quality before going to market [14–16]. Since the interspecific squash's stem is much thicker than the watermelon's (over 4 mm and about 3–4 mm, respectively), seedlings of the former species should be classified in a higher quality category in order for the stems to have relatively similar thickness.

Relatively large leaves are necessary for a greater light capture and subsequently a rapid seedling development after grafting. Seven days from grafting, the most profound differences among all tested parameters were observed in leaf area, where WH combinations, especially WH × SO, led to significantly larger leaves compared to the rest of the combinations (Figure 2, Table S1). The greater leaf area of WH combinations subsequently led to greater shoot and root dry biomass formation in the same treatments (Figure 2, Table S1). Even though the whole root system was removed during grafting, the WH × SO combination significantly enhanced the root system development in only 7 days during healing. Seedlings with vigorous root systems (greater R/S ratio) have better chances of active growth after transplanting since they have the potential to absorb adequate amounts of water and nutrients. However, grafted watermelon seedlings from all combinations exhibited a balanced shoot and root growth with no significant differences (data not shown). Faster leaf development is followed by greater photosynthetic potential, leading to increased biomass accumulation. Fourteen days after grafting, the least leaf area was found in WL and SL combinations (Figure 3, Table S2). Moreover, the least shoot dry weight was induced in the same combinations. Greater root dry weight was induced in WO × SO compared to WL × SH, and generally in WO and WH combinations (Figure 3, Table S2). Similarly to the first time interval, R/S ratio was not significantly affected by the different combinations (data not shown).

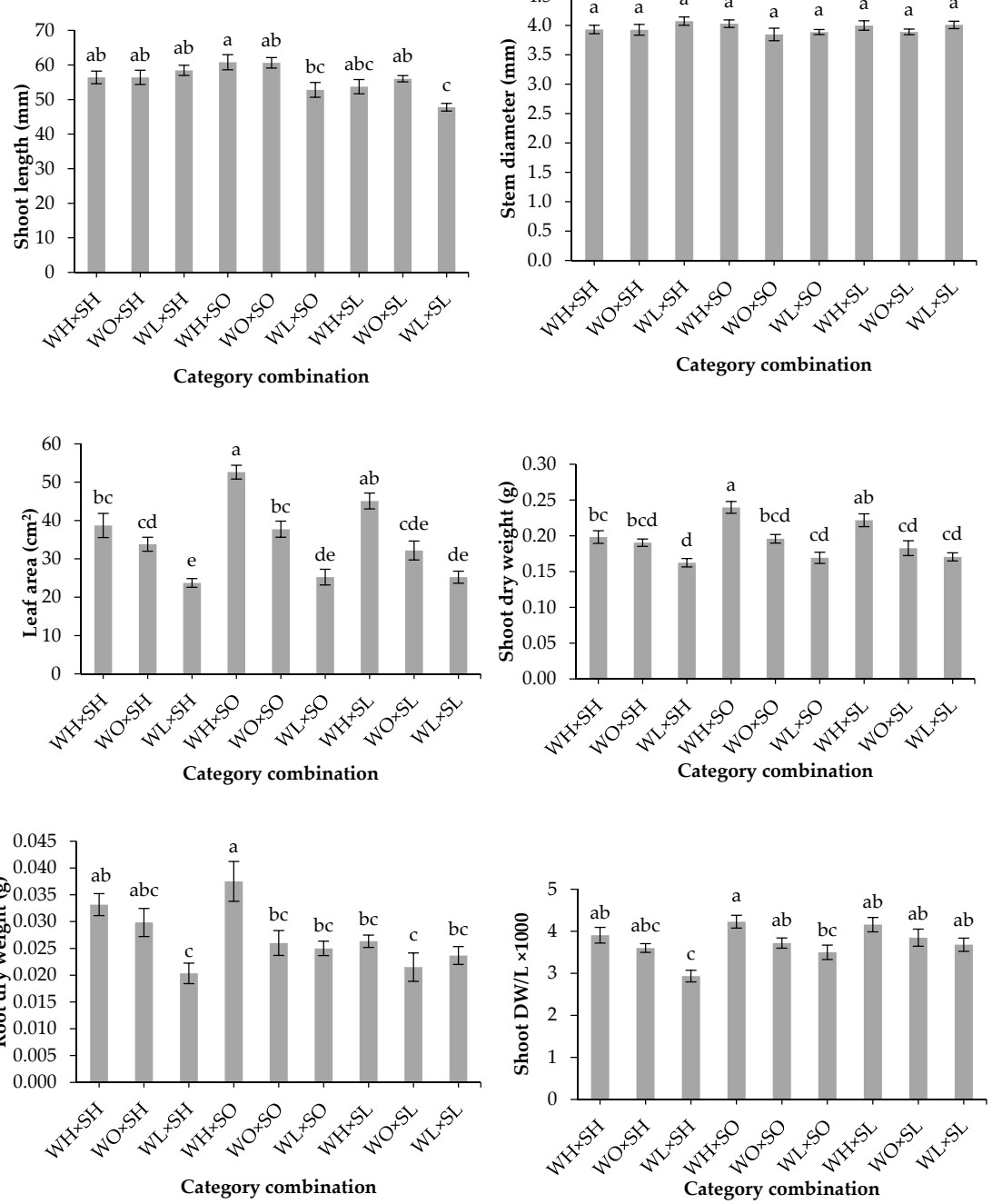

**Figure 2.** Morphological, growth, and developmental parameters of nine quality category combinations of grafted watermelon seedlings 7 days after grafting, as described in Table 1. DW/L: shoot dry weight-to-length ratio. Bars followed by different letters are significantly different ($\alpha \leq 0.05$). Each mean value was computed from $n = 12$ seedlings. Error bars correspond to the standard error (SE) of the mean.

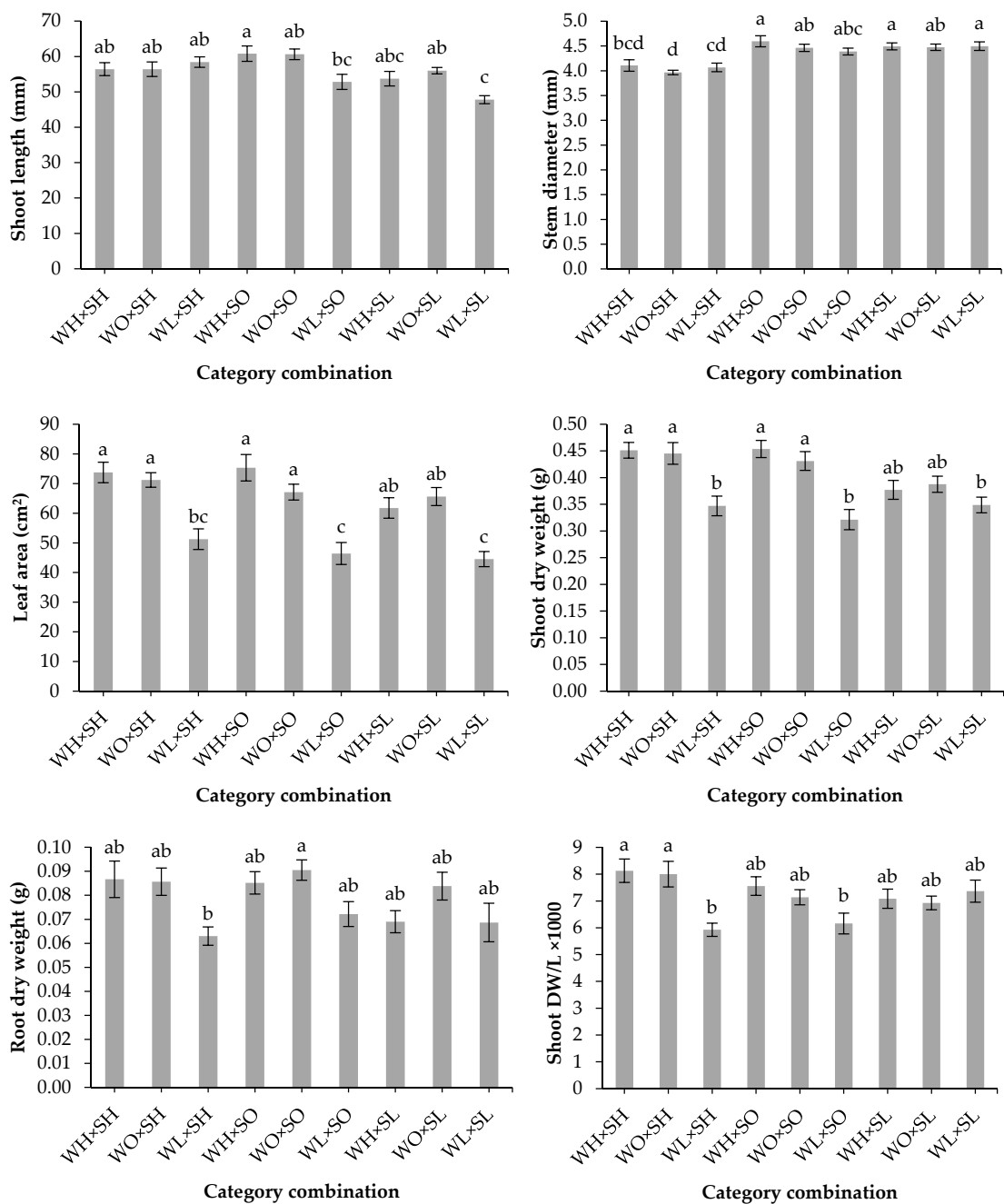

**Figure 3.** Morphological, growth, and developmental parameters of nine quality category combinations of grafted watermelon seedlings 14 days after grafting, as described in Table 1. DW/L: shoot dry weight-to-length ratio. Bars followed by different letters are significantly different ($\alpha \leq 0.05$). Each mean value was computed from $n = 12$ seedlings. Error bars correspond to the standard error (SE) of the mean.

Seven days from grafting, a seedling quality index, DW/L [1], had significantly greater values in WH × SO and generally in WH combinations (Figure 2, Table S1). Fourteen days after grafting, DW/L was promoted in WH × SH and WO × SH compared to WL × SH and WL × SO, while WL generally induced the lowest DW/L values (Figure 3, Table S2). The parameter incorporates shoot dry weight, which is decisive for the characterization of seedling quality. Therefore, at both time intervals, greater DW/L values were found in combinations with greater shoot dry weight.

In another cucurbit interaction, cucumber × pumpkin, it has been reported that scion–rootstock compatibility was characterized by the differential expression of proteins involved in photosynthesis (e.g., Rubisco large subunit), carbohydrate and energy metabolism (e.g., ATP synthase beta subunit

and ATP sulfurylase), and protein metabolism [17]. Moreover, two genes in the cucumber × pumpkin interaction, *CmRNF5* and *CmNPH3L*, have been found to be differentially expressed in compatible and incompatible scion × rootstock combinations. Specifically, the genes are related to the ABA signal pathway and the auxin transport pathway, respectively, and both genes were upregulated in compatible combinations and were proved to be stress induced [18]. These studies focused on the interaction between different hybrids of scion and rootstock, but they give an insight into what is generally happening during interspecies connections. Even though a lot of studies exist related to scion–rootstock compatibility, there is an absence of information involving the relation between the quality of to-be-grafted seedlings (scion and rootstock) and the quality of grafted watermelon seedlings.

## 4. Conclusions

A combination involving "optimum" seedlings from both species (WO × SO) was initially expected to deliver grafted seedlings of the highest quality. However, at both time intervals, a combination of watermelon "high" with interspecific squash "optimum" (WH × SO) proved beneficial for better vascular connection and the production of high-quality grafted watermelon seedlings. This combination exhibited high values in seedling quality traits, such as stem diameter, leaf area, and shoot and root dry weight, as well as DW/L. The abovementioned parameters are desirable since they proved to be valuable quality indices for grafted watermelon seedlings. Regardless of the rootstock quality, combinations including WL (12 days of watermelon scion growth) should be avoided if possible since the final product is considered low quality. The above results could help nurseries to exploit the full potential of the seedlings, as well as to provide crop growers with plant material of high quality. Future research should focus on anatomical, biochemical, and molecular aspects that contribute to the seedling production industry. Moreover, field cultivation should clarify whether the high seedling quality during transplantation is mirrored in the earliness, yield, and quality of watermelon fruits.

**Supplementary Materials:** The following are available online at http://www.mdpi.com/2077-0472/10/8/326/s1, Table S1: Morphological, growth and colorimetric parameters of grafted watermelon seedlings obtained 7 days after grafting as affected by watermelon (scion) or interspecific squash (rootstock) quality categories. Table S2: Morphological, growth and colorimetric parameters of grafted watermelon seedlings obtained 14 days after grafting as affected by watermelon (scion) or interspecific squash (rootstock) quality categories.

**Author Contributions:** Conceptualization, methodology, and data analysis: F.B., A.K., and A.S.S.; experimental measurements: F.B. and C.D.; writing—original draft preparation: F.B. and A.K.; writing—review and editing: F.B., A.K., A.S.S., and C.D.; supervision and project administration: A.K. All authors have read and agreed to the published version of the manuscript.

**Funding:** This research has been co-financed by the European Union and Greek national funds through the Operational Program Competitiveness, Entrepreneurship and Innovation, under the call RESEARCH–CREATE–INNOVATE (project code: T1EDK-00960, LEDWAR.gr).

**Conflicts of Interest:** The authors declare no conflict of interest. The founding sponsors had no role in the design of the study; in the collection, analyses, or interpretation of data; in the writing of the manuscript; or in the decision to publish the results.

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
