# Peer review of "Impact of Scion and Rootstock Seedling Quality Selection on the Vigor of Watermelon–Interspecific Squash Grafted Seedlings"

_agriculture, doi:10.3390/agriculture10080326_

Round 1

Reviewer 1 Report

In the paper entitled “Impact of scion and rootstock seedling quality selection on the quality of watermelon-interspecific squash grafted seedlings” Manuscript ID: agronomy-875827, the authors investigate the effect of different seeding qualities in the scion and rootstock on the graft quality. The main goal of this work was to study the different seeding qualities affect in the development of the graft at two stages of graft. The manuscript is novel and highly interested and the authors provide information related with the best quality during the graft to obtain high quality graft plants. Although the work seems to have been carefully carried out, I have some doubts that need to be addressed:

The title could be improved

In material and methods:

2.5 Statistical analysis: Were the data tested for normality? and for homogeneity of variances with the Levene´s test?

In results and discussion:

In Figure 1. Could you put on each graph if it is 7 or 14 days after grafting for better viewing?

In this section should provide more discussion about the results obtained with more references.

And in the last paragraph talk about genes and it should be better related to the results.

Author Response

In the paper entitled “Impact of scion and rootstock seedling quality selection on the quality of watermelon-interspecific squash grafted seedlings” Manuscript ID: agronomy-875827, the authors investigate the effect of different seeding qualities in the scion and rootstock on the graft quality. The main goal of this work was to study the different seeding qualities affect in the development of the graft at two stages of graft. The manuscript is novel and highly interested and the authors provide information related with the best quality during the graft to obtain high quality graft plants.

Although the work seems to have been carefully carried out, I have some doubts that need to be addressed:

The title could be improved

Response: Thank you for the useful comments that give us the opportunity to improve our manuscript. In-text amendments are highlighted with yellow colour.

The title was amended according to your suggestion.

In material and methods:

2.5 Statistical analysis: Were the data tested for normality? and for homogeneity of variances with the Levene´s test?

Response: Normality was determined with Kolmogorov-Smirnov test, while homogeneity of variances was determined with Levene’s test at significance level α = 0.05. This text was added in lines 160-161 in the manuscript.

Normality was not significant (p > 0.05) in almost all tested parameters and treatments.

Homogeneity of variances was not significant (p > 0.05) in all tested parameters.

In results and discussion:

In Figure 1. Could you put on each graph if it is 7 or 14 days after grafting for better viewing?

Response: Figure 1 was amended according to your suggestion.

In this section should provide more discussion about the results obtained with more references.

Response: The discussion has been improved according to your suggestion. Specifically, text and references were added in lines 190-191, 200-201, and 221-225.

And in the last paragraph talk about genes and it should be better related to the results.

Response: The last paragraph of the Results and Discussion section partly explains our finding giving a molecular aspect that would be interesting for future research. We reduced this part in order to be more consistent with the rest of the manuscript.

Reviewer 2 Report

The manuscript titled “ Impact of scion and rootstock seedling quality selection on the quality of watermelon-interspecific squash grafted seedlings” deals with the investigation of the quality of the grafting material affecting the quality of the grafted plant. The manuscript is informative, and it provides useful information for nurseries. However, the overall significance of the data are limited. It would be interesting to test the yield of the combination in the field in order to assess the significance of the potential pre-selection of the grafting material in the nursery. Finally, I would suggest to change the term “quality” with “vigour” since all the quality descriptors used are associated to the scion and rootstock vegetative growth.

Author Response

The manuscript titled “ Impact of scion and rootstock seedling quality selection on the quality of watermelon-interspecific squash grafted seedlings” deals with the investigation of the quality of the grafting material affecting the quality of the grafted plant. The manuscript is informative, and it provides useful information for nurseries. However, the overall significance of the data are limited. It would be interesting to test the yield of the combination in the field in order to assess the significance of the potential pre-selection of the grafting material in the nursery. Finally, I would suggest to change the term “quality” with “vigour” since all the quality descriptors used are associated to the scion and rootstock vegetative growth.

Response: Thank you for the useful comments that give us the opportunity to improve our manuscript.

Indeed, it is interesting to determine the yield produced by each combination. It is one of our future plans and is now stated in the Conclusions (lines 294-296) highlighted with cyan colour. However, the objectives of this study were related to the facilitation of the seedling production industry (nurseries) and thus we did not perform field experiments.

Moreover, the term “vigour” was added in the title instead of the term “quality” according to your suggestion.